# Horizontal Integration and Financing Reform of Rural Primary Care in China: A Model for Low-Resource and Remote Settings

**DOI:** 10.3390/ijerph19148356

**Published:** 2022-07-08

**Authors:** Zhi Zeng, Wenjuan Tao, Shanlong Ding, Jianlong Fang, Jin Wen, Jianhong Yao, Wei Zhang

**Affiliations:** 1Institute of Hospital Management, West China Hospital, Sichuan University, Chengdu 610041, China; 2017324020249@stu.scu.edu.cn (Z.Z.); wenjuan.tao@wchscu.cn (W.T.); huaxiwenjin@wchscu.cn (J.W.); 2World Health Organization, 1207 Geneva, Switzerland; dings@who.int; 3Chinese Center for Disease Control and Prevention, National Institute of Environmental Health, Beijing 100021, China; fangjianlong@nieh.chinacdc.cn; 4Chinese Academy of Medical Sciences & Peking Union Medical College, Beijing 100730, China

**Keywords:** integrated care, primary care, rural area, health care reform

## Abstract

Primary health care (PHC) systems are compromised by under-resourcing and inadequate governance, and fail to provide high-quality health care services in most low- and middle-income countries (LMICs). As a response to solve the problems of underfunding and understaffing, Pengshui County, an impoverished area in rural Chongqing, China, implemented a profound reform of its PHC delivery system in 2009, focusing on horizontal integration and financing mechanisms. This paper aims to present new evidence from the Pengshui model, and to assess the relevant changes over the past 10 years (2009–2018). An inductive approach was adopted, based on analysis of national and local policy documents and administrative data. From 2009 to 2018, the proportion of outpatients who sought first-contact care in rural community or township health centers increased from 29% (522,700 of 1,817,600) in 2009, to 40% (849,900 of 2,147,800) in 2018 (the national average in 2018 was 23%). Our findings suggest that many positive results have been achieved through the reform, and that innovations in financial governance and incentive mechanisms are the main driving forces behind the improvement. Pengshui County’s experience has proven to be a successful experiment, particularly in rural and low-income areas.

## 1. Introduction

Primary health care (PHC) is the most efficient and cost-effective way to achieve universal health coverage (UHC) and health-related sustainable development goals (SDGs) [1,2,3]. Efforts have been made to incorporate comprehensive PHC approaches in many countries, particularly those with strong government commitments to equity, health, and UHC. Research on primary care has also shown that health systems with stronger PHC are more likely to provide more equitable, effective, efficient, and integrated basic clinical care and public health services, and are more likely to improve health outcomes in many, but not all, high-income countries, such as England and Canada [4,5]. However, in most low- and middle-income countries (LMICs), PHC systems are compromised by under-resourcing and inadequate governance [6]. Funding for PHC is generally insufficient, access to PHC services remains inequitable, services are of inadequate quality, and health care worker shortages persist, particularly in rural areas where the need is often greatest [7]. Previous evaluations have also suggested that inequities in healthcare usage currently occur where poor populations and those living in remote areas are disadvantaged, owing to their limited geographical access to PHC [8,9,10,11]. Further, the latest COVID-19 pandemic has exposed multiple weaknesses in the health systems of both rich and poor countries [12]. Given the severe financing constraints and workforce insufficiencies in health services in many lower-income countries, especially post-COVID, attention should simultaneously be paid to both financing and service delivery enhancement [13].

The major obstacles to strengthening the PHC systems in China are the shortages in finances and workforces, especially in rural and low-resource areas [14,15]. As in many other countries, a well-rebuilt PHC system occupies a central position in China’s health care reform, which was launched in 2009. A series of comprehensive health care reforms have been advanced gradually, including improvements to the primary care delivery systems that provide basic health care services [16]. However, insufficient funds, workforce shortages, and limited multisectoral action are still regarded as potential factors impeding the full implementation of PHC [2,17]. In response to these challenges, several measures have been introduced to strengthen PHC in China, such as a three-tiered (primary, secondary, and tertiary care) health care delivery system, and a medical alliance or integrated care system. Vertical integration, by means of coordinating the availability of experienced professionals and sophisticated medical resources between higher-level hospitals and PHC services, has been greatly emphasized [18,19]. With the reform deepening, many local governments have piloted vertically integrated care models headed by tertiary hospitals, and a few studies have reported the integrated experiences of some Chinese cities, such as Shanghai [20], Luohu (Shenzhen) [21], and Tianchang (Anhui) [22]. However, evidence of horizontal integration is extremely rare; integration and coordination within PHC systems in rural China remains an important challenge with very limited empirical evidence.

Therefore, this article aims to describe the context and approaches of a horizontal integration model in rural China, to assess the relevant changes in the reforms, to discuss the lessons learned from its implementation, and to suggest what else may be required for sustainable integration. Our case study will contribute to enriching the evidence, showing how to best finance and incentivize PHC systems in order to encourage appropriate, cost-effective, equitable, and high-quality care in low-resource and remote settings. Meanwhile, the actionable evidence is realistic for rural areas in China, and our new insights into primary health care could also benefit most LMICs. Moreover, as one of the largest developing countries, China’s study could contribute to, and considerably enhance, international findings.

## 2. Methods

### 2.1. Local Setting

Pengshui County (full name: Pengshui Miao and Tujia Autonomous County), located in the remote and impoverished mountainous areas of southeastern Chongqing (Appendix A), covers an area of 3903 square kilometers and governs 3 sub-districts, 18 towns, and 18 townships. Pengshui County had a population of 498,200 as of 2016, mainly from Miao and Tujia ethnic minorities [23]. As one of the key poverty-stricken counties in China, the per capita GDP of Pengshui in 2009 was ¥10,822, less than half of the Chongqing average (¥22,920), and the annual per capita disposable income was ¥11,430, far below the provincial and national averages (¥15,749 and ¥17,175, respectively).

As one of the poorest counties in western China, the medical supply system was often plagued by insufficient local government funds and inadequate medical resources. Accordingly, there were shortages of professional medical staff and facilities, substandard treatment, frequent stockouts, sub-optimal prescription practices, and poor usage of medicines; furthermore, the payment policies did not reward quality [24]. Therefore, most residents bypassed PHC and sought health care directly from higher-level hospitals, rather than using the distrusted PHC centers closest to their homes, despite receiving a lower reimbursement of their medical expenses [25,26,27]. To solve these problems, Pengshui County implemented a profound reform of its PHC delivery system in 2009, focusing on horizontal integration and financing mechanisms [28].

### 2.2. Data Analysis

The Primary Health Care Performance Initiative (PHCPI) conceptual framework and PHC Vital Signs Profiles (VSP), a new mixed-methods assessment tool to measure PHC improvements in LMICs (Appendix A), were both adopted to guide the analysis [29,30,31]. The framework’s advantages include its abilities to build on the current health system frameworks, to describe crucial elements of a high-functioning primary healthcare system, and to provide indicators to guide efforts to enhance primary healthcare [32].

Guided by the PHCPI conceptual framework, a descriptive, explanatory, and evaluative case study was conducted, based on analysis of national and local policy documents, in-depth interviews with local policymakers and institution managers, and analysis of pre- and post-reform administrative data collected from provincial and local health administrative departments. An inductive approach was adopted for the analysis of the design and implementation of the Pengshui model, according to the governance structure, financing mechanisms, and workforce motivation, as well as service delivery. We drew on the VSP to assess the relevant improvements provided by the reform, categorized into four domains: (1) Financing, to measure PHC financing prioritization; (2) Capacity, to assess functional capacity, including governance, workforce, inputs, and facilities; (3) Performance, which focuses on service delivery; and (4) Equity, which highlights equity in access and service coverage.

The data were collected between 2009 and 2018 from the health administrative departments in Chongqing, with the local health departments’ permission, and from the Yearbook of Health Statistics, and different sources of material were used to gather evidence of these experiences.

## 3. Results

Pengshui County integrates the PHC resources which are driven by the government’s “funds pool” to create a horizontally integrated primary healthcare institution group model (Pengshui model). It aims to improve the capacity of PHC services, to promote the long-term sustainable development of PHC institutions, and to solve the problems of accessibility and affordability of medical treatment for residents.

### 3.1. Implementation

#### 3.1.1. Governance Structure

Since 2009, 36 township and 4 community health service centers in Pengshui County have been organized into a “horizontal compact medical consortium”—the primary health care institution group (PHCIG). The governance structure (layered jurisdiction) includes: (1) Decision-making: the Commission of Primary Health Care (CPHC) was established to manage the group, it is responsible for the group’s development and major investment planning, and is mainly composed of the leaders and members of the local government health department; (2) Management: the Rural Health Management Center (RHMC) (including the Health Accounting and the Centralized Drug Purchasing Centers) was established to assume responsibility for the management of strategic planning, the funding pool and performance, centralized drug procurement, etc.; and (3) Implementation: the director of each primary health care institution was made responsible for the internal business management. The Pengshui Model’s overall governance structure diagram (Figure 1) indicates the relationships between local government, PHC management organizations, and PHC institutions. It also highlights crucial PHC elements, including political commitment and leadership, financing and strategic purchasing, resource and workforce allocation, management of service delivery, and performance monitoring.

#### 3.1.2. Financing Mechanisms

A creative financial method based on the PHC-Funds was introduced. As long-term accumulation funds, they are composed of government subsidies and funds raised by 40 PHC providers (Figure 2). The funds were targeted for infrastructure construction, equipment purchasing, talent training, and financial incentives, including guaranteed minimum salaries. In addition to the strategic investment at the group level, each PHC institution could also apply for funds according to its own development needs. As with interest-free repayments, excess funds should be refunded in full without interest. In addition, multiple financial regulations were provided to guarantee that the funds were utilized efficiently and rationally.

#### 3.1.3. Workforce Motivation

Human resources reform focusing on talent attraction and talent training was implemented with the support of the PHC-Funds via payment incentives. The PHCIG used a staff pooling strategy to recruit, train, and distribute medical professionals into appropriate positions as needed, enhancing the flexibility and availability of human resources. The PHC workforce salary system was enhanced, and a performance-based payment was introduced. Therefore, doctors, pharmacists, nurses, and public health personnel received monthly salaries and quarterly premiums. Medical staff were encouraged to undertake advanced training at higher teaching hospitals, and extra subsidies were paid to physicians receiving on-the-job training in secondary and tertiary hospitals (¥10,000–60,000 annually). Additionally, a guaranteed minimum wage (approximately equal to the first class) and an additional bonus (¥600–800 per month) were provided for qualified health professionals working in outlying institutions.

#### 3.1.4. Service Delivery

From 2009 to 2018, 40 basic-level medical institutions were rebuilt or expanded, creating a hubs-and-spokes network, and the three-level healthcare system was improved at village, township, and county levels. First, 12 hubs were strengthened, where the CPHC assembled talented staff and housed sophisticated equipment in densely populated areas. Then, 28 spokes were added one-by-one in outlying villages and towns, focusing mainly on public health care, routine treatment, and follow-up care, while sending those requiring higher-level diagnoses and treatment to the nearest hub or secondary/tertiary hospitals. This model helped with accessibility for the underserved population, and guided all patients to the appropriate level of care.

### 3.2. Achievements

To enable a more explicit examination, and to make it more adaptable to the Chinese context, we incorporated a Chinese version of the UHC indices within the PHCPI framework, in order to analyze the changing trends in affordability and accessibility over the last 10 years (2009–2018) in Pengshui County. These two aspects reflect the challenges in the Chinese health system well, namely, “difficulty in seeking medical services” and “high expenses related to medical service utilization”, and are also in line with the UHC dimensions of “essential health services coverage” and “financial risk protection”, as defined by the WHO [14]. This refined index conceptualizes the idea that the structural and process dimensions of a PHC system contribute to four PHC outcomes: financing and equity, reflecting affordability; and capacity and performance, reflecting accessibility (Table 1).

#### 3.2.1. Financing

Since the establishment of the PHC-Funds in 2012, more than 400 million yuan has been raised. Meanwhile, local government investment in PHC has been increasing year by year, with an average annual growth rate of almost 30%. From 2009 to 2018, the percentage of government health expenditure on PHC increased from 14.35% to 53.14%, and the mean annual payment to primary care staff increased to ¥120,800 (SD ¥25,615 per year), which was five times higher than that in 2009 (the national per capita disposable income of residents was ¥28,228 in 2018).

#### 3.2.2. Equity

PHC systems can effectively serve the most marginalized and disadvantaged groups in society, improving the health of the entire population by targeting resources where needs are the greatest [30]. With the help of government subsidies, coverage of basic medical insurance for the poor has reached over 99.9%. Initiatives were taken to re-subsidize the medical expenses of poor patients, and to fully implement poverty alleviation measures such as “diagnosis and treatment first, pay later” and “one-stop” express settlement services for poor patients. Through these measures, all poverty-stricken populations can now enjoy basic medical insurance, critical illness insurance, and medical assistance, ensuring that the poor can obtain medical treatment for major serious illnesses, and guaranteeing medical services for critical illnesses. Since 2015, a total of 139 million yuan have been distributed from health poverty alleviation funds, benefiting 349,700 patient visits.

#### 3.2.3. Capacity

The most significant parts of the “capacity” dimension, which reflects the ability to deliver high-quality PHC, are medications and supplies, facility infrastructure, information systems, and manpower. In comparison to before the reform, overall, medical facility workplaces have expanded 7.2 times, and the number of medical beds has tripled to 1242. With the help of PHC Funds, 318 rural clinics have been reconstructed, and 33 mobile clinics have been established to serve villagers living in remote mountainous areas. Furthermore, advances in digitization and internet capabilities have made it far easier for PHC doctors to seek assistance from higher-level hospitals.

#### 3.2.4. Performance

Notably, the increase in financial and human resources has encouraged people to meet their health needs and use more outpatient and inpatient services at the PHC level (Figure 3). From 2009 to 2018, the proportion of outpatients who sought first-contact care in rural community or township health centers increased from 29% (522,700 of 1,817,600) in 2009 to 40% (849,900 of 2,147,800) in 2018 (the national average in 2018 was 23%). However, the number of inpatients in PHC nationwide has been declining in the same period. Although the proportion of inpatients in primary health institutions in Pengshui County decreased slightly from 67% (30,528 of 45,450) to 60% (93,273 of 156,166), it is worth noting that the number of inpatients tripled, and that the declining trend has been slowing since 2015 at a rate that is 6 percentage points lower than the national average (the national average has decreased from 31% to 17%), indicating that PHC centers are slowly regaining their appeal.

## 4. Discussion

### 4.1. Lessons Learned

The question of how best to strengthen PHC, especially in terms of health financing and human resources, has been a closely studied topic for decades, with a focus on integrated care networks. Vertical integrations of the healthcare system in China have resulted in improved professional competency, better care coordination, and a stronger capacity to satisfy patients’ needs, while the positive impacts have varied depending on the type of integration [18,33]. Evidence from a longitudinal study suggests that vertical integration (especially tight integration) in China has significantly contributed to strengthening primary healthcare in terms of inpatient services and quality in hypertension and diabetes care [33]. Higher-level hospitals play a dominant role in the interprofessional collaboration, particularly in allocating more healthcare resources to PHC, assisting PHC institutions set up clinical departments with high demands, and conducting joint technical training [34]. Comparatively, under loose collaboration, the integration strategies between participating hospitals and PHC mostly focus on the service level, frequently involving technical help and skills training, and very seldom occur at the organizational or administrative level [33]. Notably, the funding mechanism of PHC has not altered; in both tight and loose vertical integration models, the PHC institutions are still financed by local governments. Moreover, vertical integration (particularly tight integration) has increased the possibility of the most capable PHC professionals flowing to the hospitals in search of better compensation and career satisfaction, resulting in a greater loss of personnel in PHC [18].

In comparison to some other typical integrated care models in China (Table 2), three major lessons learned from the strengthening of the primary health system through horizontal integration are worth highlighting.

First, adequate and long-term funding is critical to the development of a strong PHC system [36]. Our case study suggests that a top priority of Pengshui County’s reform strategy is to mobilize additional pooled funding and sufficient resources to fund the sustainable and equitable provision of PHC. The policy of compulsory payment for fund-raising and mutual aid-sharing has improved the financing and compensation mechanisms, and made up for the problem of insufficient investment in underdeveloped counties. This pooled funding strategy is an innovative endeavor to attain common prosperity in health care, and it has helped to achieve coverage of health services in remote places while continuing to meet the needs of the poor population. Equity enhancing and resource allocation mechanisms should be developed to distribute financial and other resources to the poor and remote areas where they are most needed, in order to reduce poverty and attain internationally agreed development goals, including SDGs [37].

Second, addressing the PHC workforce gap to ensure adequate and well-trained medical staff is an immutable priority when strengthening PHC [38]. The Pengshui model is instructive in its illustration of the strong reliance on building a sustainable primary care workforce with a complementary financing system to create the needed incentives for productivity as well as accountability [39]. In this case, they used health education policies to retrain specialists to become general practitioners, helping to staff rural and underserved areas in the country [37]. A reasonable salary incentive system helps increase the attractiveness of staying in outlying areas to PHC professionals. Compared with the interim technical guidance and assistance from the vertical medical alliance, it is more conducive to encouraging local practitioners to receive longer and continuous training at higher-level hospitals, in order to improve targeted and practical capabilities and skills, so as to radically change the long-term status of insufficient or “low-quality” PHC services [40,41].

Third, strengthening health systems with a PHC-based foundation demands savvy political leadership and sustained commitment, as well as proactive, adaptable financing arrangements to engage stakeholders at all levels, while taking into consideration the political, social, and economic contexts [13]. Most existing vertical alliances in China are loose networks that do not have shared responsibilities, management, patient care, or economic interests across member facilities [34]. In comparison, this horizontal model has established a clear division of responsibilities and provides a governance structure with a long-term vision for PHC. The governing power of PHC institutions, previously dispersed among different departments, is consolidated into one commission. In addition, empowering PHC teams and communities helps to improve participation in decision-making, autonomy, responsiveness to population health needs, and local accountability [37]. These principles can help local governments build coalitions of supportive stakeholders and create momentum when transitioning to innovative, comprehensive, people-centered, and coordinated PHC models.

Additionally, strengthening PHC through an integrated approach is essential for building a resilient health system. In response to the COVID-19 epidemic in Pengshui County, PHC played a gatekeeping role in identifying and triaging suspected cases, making early diagnoses, and supporting the provision of vaccination services. Evidence gathered since the start of the pandemic has shown that the collaborative medical service networks have significantly improved the preparedness and resilience of the health care services, including mobilizing additional financing from PHC-Funds, coordinating medical facilities and supplies, reinforcing the availability of health workers, and, most importantly, maintaining delivery of essential health services in cases where larger hospitals closed their outpatient departments during periods of transmission. Similarly, a study of the Luohu model in Shenzhen, China, supported the finding that the core strategies and mechanisms of an integrated healthcare system can contribute to improving the public health capacity during an emergency response [42]. The COVID-19 pandemic is revealing serious gaps and vulnerabilities in the health systems of many countries; hence, accessing high-quality PHC is as important as ever [13]. We need to take appropriate action to support the role of primary care by building and rebuilding resilient health systems [43].

### 4.2. Future Challenges and Next Steps

The Healthy China Strategy sets out principles for improving the quality of primary care delivery in poor settings and emphasizes the requirement for greater integration of PHC. To achieve this goal, the essential values of universal access to care, equity, community engagement, intersectoral collaboration, and resource allocation must be all followed.

However, despite substantial financial investment and infrastructure building in the past decade, the PHC system is still facing challenges in providing care of high quality and high value [26,35,44]. As governments increase health expenditure, there is an urgent need for value-based payment mechanisms to reinforce provider accountability for population-wide health outcomes, including for poor and vulnerable people. Examples include combining the public health budget with the social health insurance budget, and shifting the payment of PHC teams from a fee-for-service regime to a capitation payment method based on prepayment, pooling, and strategic purchasing [13,45]. By doing so, this incentivizing payment will encourage PHC physicians to better coordinate preventative care with clinical care. As well as improving drug accessibility for poor patients and lowering their treatment financial burden, the government’s centralized drug procurement should be further expanded; a reasonable range of medications, more technologies, and innovative medicines should be added to meet the people’s demand for high quality [46]. Moreover, simply adding more health workers will not suffice, as a strong foundation for trusting relationships within the workforce needs to be strengthened, supported by high-quality training, strong and effective supervision, and appropriate compensation [47]. To achieve this, team-based approaches could be considered a promising way to scale quality, patient-centered primary care services; inclusive of family doctors and community health workers being actively engaged with patients and their families in healthcare delivery [48]. Furthermore, good continuity and coordination of care are critically important. Both the vertical and horizontal approaches should be combined to create a more powerful PHC system [49]. The coordination between primary healthcare services and hospitals, and the engagement of the private sector in PHC are therefore priorities for reorienting health services to the needs of the people, particularly for LMICs with shortages of health care workers and many dispersed, remote communities [50].

### 4.3. Study Strengths and Limitations

To our knowledge, this study is the first to provide insight into the horizontally integrated system reform of rural primary care, anchored in real-world situations in China. The case study results in a rich and holistic account of a phenomenon, providing new evidence of the effects of primary care reform over ten years, and the policy implications for similar settings. However, data accessibility is essential for evaluating PHC performance, particularly patient experience and care quality. Additional performance assessment tools, such as composite indices and national performance dashboards, should be developed and tested by the PHCPI [29]. Additionally, VSP can be impacted by a variety of variables, including age, season, gender, medication, and the effects of the environment. Healthcare professionals need to be aware of the many physiological and pathologic processes that affect these collections of measurements, and how to interpret them correctly. As has been discussed, this is only a single descriptive case study due to the limited available data; further detailed research requires comprehensive data collection and multiple analyses to consolidate our findings.

## 5. Conclusions

As evidenced over the past 10 years, many accomplishments have been achieved through the Pengshui Model, including advanced medical facilities, an increase in the number of qualified medical professionals, better PHC capability, and improved patient trust in public health and primary medical services. The innovation of the financial governance and incentive mechanism is the main driving force for the improvement. The Pengshui Model has already become a regional policy, with scaled implementations in Chongqing’s other 25 districts and counties from 2019. Overall, Pengshui County’s experience sets an example in building integrated care delivery systems at a local government level, not only for rural areas in China, but also in other rural and low-income areas confronting similar challenges.

## Figures and Tables

**Figure 1 ijerph-19-08356-f001:**
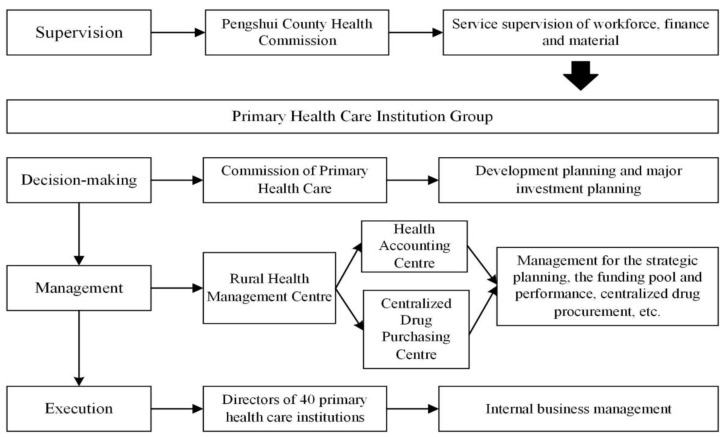
The governance structure of the Pengshui model.

**Figure 2 ijerph-19-08356-f002:**
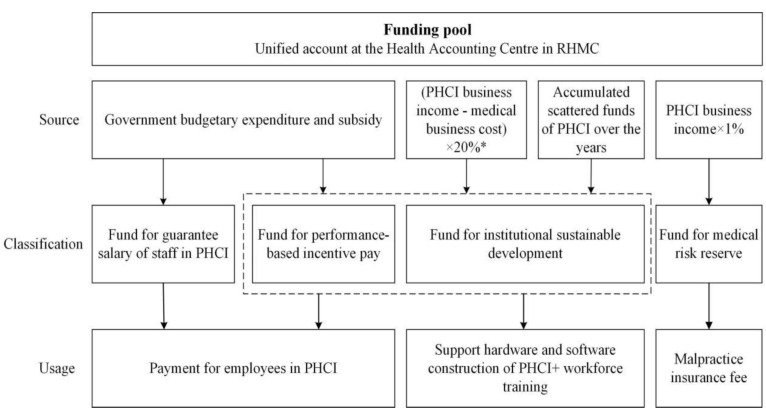
The source, classification, and usage of the funding pool in Pengshui County. * 20%: 10% is the percentage of the income from medical services (without pharmaceuticals), and another 10% is the percentage of income including pharmaceuticals. PHCI = Primary Health Care Institution, RHMC = Rural Health Management Centre.

**Figure 3 ijerph-19-08356-f003:**
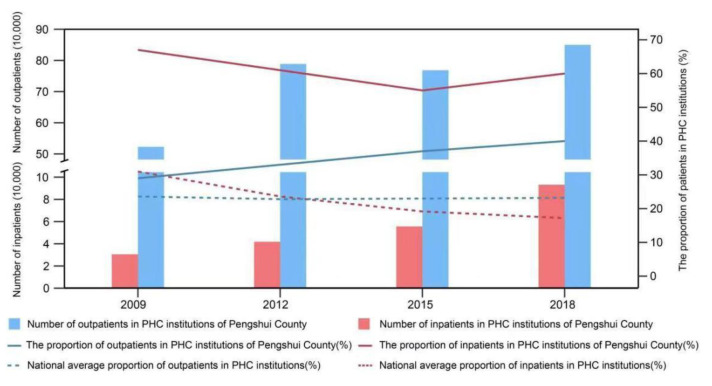
Comparison of visits to Pengshui’s PHC institutions, 2009–2018. Source: Health Statistics Yearbook. The proportion of outpatients in PHC institutions (%) = number of patients who sought first-contact care in PHC institutions/number of patients in healthcare institutions at all levels. Primary health care = PHC.

**Table 1 ijerph-19-08356-t001:** Relevant changes in Pengshui County primary health care reform over ten years (2009–2018).

Index	Domains	Indicator	Definition	Year		% Change (2009–2018)
				2009	2018	
Affordability	Financing	Government spending on PHC as % of government health spending (%)	Share of general domestic government health expenditure allocated to PHC	14.35	53.15	+270.38
		Total PHC spending per capita per year (¥)	The absolute amount of spending on PHC per person per year	86.41	429.44	+396.98
	Equity	Average cost per outpatient in PHC (¥)	Patients’ average financial burden	38	69	+81.58
		Average cost per inpatient in PHC (¥)	Patients’ average financial burden	819	1615	+97.19
Accessibility	Capacity	Number of medical beds in PHC	Availability of health facility	410	1242	+202.93
		Number of physicians in PHC	Availability of health workforce	363	857	+136.09
		Number of nurses in PHC	Availability of health workforce	66	209	+216.67
		Number of health technicians per 1000 population	Availability of health workforce	1.61	5.14	+219.25
		Number of physicians per 1000 population	Availability of health workforce	0.73	3.12	+327.40
		Number of nurses per 1000 population	Availability of health workforce	0.43	2.12	+393.02
	Performance	Number of outpatient visits to PHC (10,000)	Utilization of outpatient services	52.27	84.99	+62.60
		Number of inpatients in PHC (10,000)	Utilization of inpatient services	3.05	9.32	+205.57
		% of outpatient service utilization at PHC level (%)	The attractiveness to patients to use PHC services	29	40	+37.93
		% of inpatient service utilization at PHC level (%)	The attractiveness to patients to use PHC services	67	60	−10.45

Source: Health Statistics Yearbook and health system reform surveillance data. Primary Health Care = PHC.

**Table 2 ijerph-19-08356-t002:** Comparison of typical integrated care models in primary care in China.

Typical Model	Context (2020)	Type of Integration	Involved Providers	Funding and Incentives Mechanism	Governance Structure
Pengshui	Location: county, in a rural area Population: 530,599 GDP: 24.51 billion Area: 3903 km^2^	Horizontal integration	Township health centers and community health centers	The policy of fundraising and mutual aid-sharing of compulsory payment improved the compensation mechanism	A clear division of responsibilities and governance structure was developed by establishing a primary health care institution group
Luohu	Location: district, in an urban area Population: 1,143,800 GDP: 237.53 billion Area: 78.75 km^2^	Vertical integration	Community health centers and local hospitals	A needs-based capitation approach in social health insurance reimbursement, accompanied by differentiated pricing policies, to incentivise primary care groups to save costs	A primary care group was established, which is a network of integrated management, shared responsibilities, and common interests
Tianchang	Location: county, close to urban area Population: 603,780 GDP: 54.93 billion Area: 1770 km^2^	Vertical integration	Township health centers, community health centers, and local hospitals	An incentive distribution mechanism was designed to contain costs by providing more preventive care to residents	Three primary care groups were established, two of them were led by local public hospitals, and the other group’s leading hospital was a non-profit private hospital

Source: the data in the context come from the seventh National Census and local statistical yearbook. Luohu and Tianchang models refer to various studies, including those of Xin Wang et al. [21], Weilong Lin et al. [22], and Xi Li [35] et al.

## Data Availability

The data presented in this study are available on request.

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
