# Peer review of "Horizontal Integration and Financing Reform of Rural Primary Care in China: A Model for Low-Resource and Remote Settings"

_ijerph, 2022, doi:10.3390/ijerph19148356_

Round 1
Reviewer 1 Report
Summary of the Work
Shortages of finance and workforce, especially in rural and low resource areas, are the main obstacles to strengthening the primary health care (PHC) systems in China. Based on the analysis of national and local policy documents, in interviews with local policymakers and institution managers, and administrative pre-reform and post-reform data, the authors investigated the effectiveness of a reform of the PHC delivery system implemented by the Pengshui Miao and Tujia Autonomous County in 2009.
Main Results Obtained
The Pengshui model is based on horizontal integration and funding mechanisms.
i) The Pengshui Model allowed
- the achievement of advanced medical facilities;
- an increase in the number of qualified medical professionals;
- a better capability of PHC;
- to get an improved patients’ trust in public health and primary medical services.
ii) The innovation of financial governance and incentive mechanism is the main driving force for the improvement;
General Remarks
- The objective of the work is well motivated even if many sentences are repeated several times in the manuscript.
- The governance structure of the Pengshui Model as well as the source, classification, and usage of the funding pool in Pengshui County are described in a clear way.
- The data analysis follows the methodology established by the Primary Health Care Vital Signs Profiles according to the standard domains: Financing, Equity, Capacity and Performance.
- The description of the methods adopted is clear. However, the description of the categories and the indicators that have been chosen as important measures of domains (the vital signs domains) for PHC needs further clarifications.
- The statistical method adopted to obtain the conclusive results shown in Figure 3. also needs further clarification.
Suggestions
The authors decided to adopt the Vital Sign Profile (VSP) methodology to perform their analysis. In my opinion, in this case the VSP protocol have to be followed to the letter and executed in its entirety. So, I suggest the authors to make an extra-effort and re-organise the data illustrated in the Table 1. according the following (standard) VSP-scheme. This exercise will facilitate the reading of the work and will allow to evaluate the VSP-scores (see suggestion 5) below).
1) In the domain "Equity" please specify the following sub-indicators for the years 2019 and 2018 (according to the standard definition given by the VSP methodology):
- Equity in access
- Equity in coverage
- Equity in outcomes
2) Please, specify the "Performance" sub-indicators for the years 2019 and 2018 according to the three key dimensions of service delivery and the definition given by the VSP methodology:
- Access
- Quality
- Coverage
3) Please, specify the "Capacity" sub-indicators for the years 2019 and 2018 according to the three key dimensions of service delivery and the definition given by the VSP methodology:
- Governance
- Inputs
- Population health & facility management
4) Finally, please re-organise the indicators reported in Table 1. concerning the “Financing” in terms of the following sub-indicators for the years 2019 and 2018:
- Total spending on PHC (current PHC Expenditure per capita)
- Prioritisation of spending on PHC (current PHC expenditure as % of current health expenditure and domestic general government PHC expenditure as % current PHC expenditure)
- Sources of spending on PHC
For clarity, I am not asking to delete Table 1., which expresses the changes in the Pengshui County primary health care reform for ten years (2009-2018) according to the "Affordability" and "Accessibility" indices. This table is very informative and should be shown. I just ask to add, if possible, another table built according to the above-suggestions 1) - 4).
5) Based on the results obtained by answering suggestions 1)-4), please calculate the sub-domain scores and show the scores that will appear on the VSP.
6) The statistical analysis used to obtain Figure 3 is unclear. Please,
6a) Show the magnitude of the errors of the experimental data;
6b) Explain the meaning of the straight lines. Are the straight lines the result of linear regression analysis? If this is the case, as known, in the linear regression analysis it is assumed that the cause-and-effect relationship between the variables remains unchanged. This is a strong assumption and this hypothesis may not hold good during the evolution and, hence, estimation of the values of a variable made on the basis of the regression equation may lead to erroneous and misleading results.
The authors are asked to clarify the above points.
7) The authors adopted the PHCPI conceptual framework and PHC-VSP to guide our analyses. Indeed, this is a very good approach. However, it is useful to mention to the reader what are the limitations of these methods. For instance, I am just mentioning some of them), the availability of data is critical to assessing PHC performance, particularly patient experience and quality of care. The PHCPI have to continue to develop and test additional performance assessment instruments, including composite indices and national performance dashboards. In addition, VSP can be influenced by a number of factors. It can vary based on age, time, gender, medication, or a result of the environment. Healthcare providers must understand the various physiologic and pathologic processes affecting these sets of measurements and their proper interpretation. The authors are invited to provide a comment in this respect.
Conclusions
The work is interesting and I enjoyed reading it. However, I think that to attract the reader more it is necessary to leave the local context (the Pengshui county). By this, I mean that, of course, we have to study the Pengshui model as a special case. However, this in order to set up successively a general approach which applies also for other counties. The aim of the suggestions expressed above is precisely that of proposing to reformulate the work in general terms, by inserting a supplementary table according to the standards foreseen by the PHC-VSP methodology. So, I encourage the authors to take my suggestions into account; in my opinion this will help improve the scientific soundness of the work.
Author Response
Dear Editor,
We appreciate you and the reviewers for your precious time in reviewing our paper and providing valuable comments. It was your valuable and insightful comments that led to possible improvements in the current version. The authors have carefully considered the comments and tried our best to address every one of them. We hope the manuscript after careful revisions will meet your high standards. The authors welcome further constructive comments if any. Below we provide the point-by-point responses. All modifications in the manuscript have been highlighted in red.
Sincerely,
Wei Zhang, M.D.
Professor, West China Hospital, Sichuan University
And
Jianhong Yao, Ph.D.
Professor, Chinese Academy of Medical Sciences & Peking Union Medical College

Reviewer 2 Report
This paper was based on a single case study about the primary health care reform and assessed the effectiveness of a novel and horizontal cared model in an improved area in China. However, the overall manuscript may need considerable improvement in terms of structure and explanations.
1. Introduction
From line 43-45, The authors mentioned “Previous evaluations also suggested that the inequity of healthcare usage currently occurs, where poor populations and those living in remote areas would benefit less due to their limited geographical access to PHC”. The author only provided one citation here to support this point, which is not sufficient to show how serious and popular this issue is in China. More evidence should be provided here.
Line 45-46, “the latest COVID-19 wave” is not good wording. The author could use pandemic to replace wave or just remove wave. Also, the followed-up sentence “Exposed a long-term problem of underfunded and understaffed PHC”. What long-term problem that is particularly brought up by COVID-19 for the PHC in China? This should be further explained
Line 58-64, what is the meaning of “tertiary hospitals”? Also, the author mentioned both “vertical integrated care model” vs “horizontal integrated care model”, what are their essential differences, and why they were proposed in this article. The author should provide more background on this point.
Line 68-71, the author mentioned “Our case will contribute to enriching the evidence on how to best finance and incentivize PHC systems to encourage appropriate, cost-effective, equitable, and high-quality care in low-resource and remote settings, and mean-while, set an example not only for rural areas in China but also for other LMICs”. Why is this case study in the Chinese context informative for other low-income and middle-income countries? This should be further explained.
2. Method.
Data analysis
In Line 94, “Guided by the Primary Healthcare Performance Initiative (PHCPI) conceptual 94 frameworks”. The conceptual framework should be mentioned and further explained in the introduction section.
line 101-103 “Relevant improvements of the reform were assessed in four domains: financing and equity reflecting affordability, capacity, and performance reflecting accessibility. What exact index under each domain should be assessed? These should be further explained to inform the readers.
3. Result
Figure 1 and Figure 2 look not clear. Maybe it is about the resolution of the picture. Also for the government structure, how to assess this newly established structure in Pengshui County? How was it related to the previously mentioned four domains?
For figure 2, line 141-142, “one 10%” may be a typo, but how do these numbers correspond to picture 2.
Line 147, what is “A staff pooling strategies” meant here? The author should give more detailed information here.
Line 170, the author mentioned, “we incorporated Liu’s analysis indices within the PHCPI framework”. What is Liu’s indices? This should be introduced in the analysis section.
For table 1, the numbers for 2009 and 2019 are unclear. What are the numbers here refereeing? What is the unit for these numbers?
4. Discussion
Line 226-228, “Evidence from a longitudinal study suggests that vertical integration (especially tight integration) in China significantly contributed to strengthening
primary healthcare in terms of inpatient services and quality of hypertension and diabetes”. Still, what is the vertical integration model? Why comparing the vertical vs. horizontal integration model is important for this study?
The author mentioned in-depth interview data in their analysis section. But I did not see there was any discussion based on the interview data. The authors could consider removing it.
Author Response

(The authors gave the same response as above.)

Reviewer 3 Report
The lack of resources to invest in Primary health care (PHC) systems has been a problem in many parts of the world. However, as far as is known, many governments in several countries are not yet committed to universal access to health, especially for the low-income population. In this study, the authors show that Pengshui County, an impoverished area in rural Chongqing, China, implemented a profound reform of its PHC delivery system in 2009, focusing on horizontal integration and financing mechanisms.
The authors show that many positive results have been achieved through the reform, and the innovation of financial governance and incentive mechanism is the main driving force for the improvement. In this context, Pengshui County's experience has proven to be a successful experiment, particularly in rural and low-income areas.
I congratulate the authors for their research, article structure, and data analysis. But, on the other hand, some crucial issues need to be highlighted in the manuscript:
1 - The first point is related to the relevance of the study. How could other regions in China and the world benefit from this study?
2 - Why was the analysis carried out from 2009 to 2018? Could the current pandemic period change the results found? Discuss this point.
3 - "Pengshui County integrates PHC resources which are driven by the government's "funds pool" to create a horizontally integrated primary healthcare institution group model (Pengshui Model). It aims to improve the capacity of PHC services, promote the long-term sustainable development of PHC institutions, and solve the problem of accessibility and affordability of medical treatment for residents". What is the government's primary role in solving the problem of the population's access to the most expensive treatments? Like new technologies and innovative medicines?
4 - The authors carried out a descriptive, explanatory, evaluative case study based on an analysis of national and local policy documents, in-depth interviews with local policymakers and institution managers, and analysis of pre-reform and post-reform administrative data from provincial and local health administrative departments. There are no details on the methodology of the sample selected for the interview. How was the interview conducted? Was a structured questionnaire applied? Was the interview recorded? What questions were asked? I did not observe the ethics consent form in the files.
5 - Improve the quality of Figures.
6 - Figure 3 shows that the proportion of inpatients in primary health institutions decreased, indicating that more patients choose hospital care rather than PHC. However, the declining trend began slowing in 2015, marking a decline rate six percentage points lower than the national average. From a practical and managerial point of view, does this mean an improvement? Justify your answer.
7 - Compared to some typically integrated care models in China (Table 2), three major lessons learned are worth highlighting from strengthening the primary health system through horizontal integration. In this context, what lessons can research show from vertical integration?
8 - To the best of our knowledge, governments increase health expenditures in many regions. There is an urgent need to combine the public health budget with the social health insurance budget and shift the payment of PHC teams from fee-for-service to a capitation payment method based on prepayment, pooling, and strategic purchasing. In this context, how can the topic of VBHC (Value-Based Health Care) be explored?
9- A weak point of work is related to a single simple case study without allowing to perform triangulation and make generalizations. More comprehensive data collection and other analyses, including statistics, are needed to consolidate the findings for greater robustness.
Author Response

(The authors gave the same response as above.)

Round 2
Reviewer 1 Report
The authors answered all the questions raised in my previous report improving the quality of the manuscript significantly. Yes sure, there are still other points that could be improve. However, in my opinion this version of the manuscript may be published.
Reviewer 2 Report
The authors sincerely addressed my questions and improved the quality of the manuscript.
Reviewer 3 Report
Dear authors,
Thank you very much for responding point by point to what was suggested. The article got better and has the potential to be published.